# Perception of Environmental Risks and Behavioral Changes during Pregnancy: A Cross-Sectional Study of French Postpartum Women

**DOI:** 10.3390/ijerph16040565

**Published:** 2019-02-16

**Authors:** Raphaëlle Teysseire, Marion Lecourt, Jim Canet, Guyguy Manangama, Loïc Sentilhes, Fleur Delva

**Affiliations:** 1Environmental Health Platform Dedicated to Reproduction, ARTEMIS Center, 33076 Bordeaux, France; guyguy.manangama@chu-bordeaux.fr (G.M.); loic.sentilhes@chu-bordeaux.fr (L.S.); fleur.delva@chu-bordeaux.fr (F.D.); 2Department of Occupational Medicine, Bordeaux Hospital, Bordeaux, 33076 Bordeaux, France; 3Midwifery School, 33076 Bordeaux, France; lecourtmarion00@gmail.com; 4Bordeaux Population Health Research Center, Inserm UMR1219-EPICENE, University of Bordeaux, 33076 Bordeaux, France; jim.canet@u-bordeaux.fr; 5Department of Obstetrics and Gynecology, Bordeaux Hospital, Bordeaux, 33076 Bordeaux, France

**Keywords:** environmental health, pregnancy, prenatal care, prevention, behavioral changes, risk perception, occupational health, health knowledge, attitudes, practice

## Abstract

Limiting exposure to environmental hazards during preconception and pregnancy is essential for preventing adverse pregnancy outcomes or developmental defects in offspring. However, the perception of environmental risk and the behavioral changes of women planning or having a pregnancy have rarely been investigated, except for a few risk factors. We thus performed a cross-sectional study of French postpartum women hospitalized in the Bordeaux University Hospital in 2017 by proposing a self-administrated survey. The main objective was to assess their level of awareness concerning a large panel of environmental hazards and modifications in their behavior during pregnancy in occupational and household environments. Among the 121 respondents, most identified the environment as a major factor for a healthy pregnancy but recognized a lack of knowledge regarding environmental risk factors. The internet, television, and magazines were their main sources of information. Most women modified some of their practices at work or home. These measures were rarely implemented in consultation with a health practitioner, which raises concerns about the relevance of the adjustments made. Our findings highlight the need to improve the quality of information available to women and to help them implement preventive measures in consultation with physicians.

## 1. Introduction

Environmental exposure to reproductive hazards during the periods of conception, pregnancy, and breastfeeding can lead to adverse pregnancy outcomes or developmental defects in offspring. The lifestyle of women before and during pregnancy is a crucial component of fetal and child development. Several studies have shown that prenatal consumption of tobacco, alcohol, or cannabis is associated with reduced birth weight and preterm delivery, as well as behavioral and cognitive deficits during childhood [1]. Recent epidemiological studies have raised concerns about prenatal exposure to substances found in daily products, such as brominated flame retardants, phthalates, and pyrethroids, which may cause adverse cognitive and behavioral outcomes in offspring [2]. Finally, several activities, such as home renovation, can lead to exposure to lead or organic solvents, both associated with adverse pregnancy outcomes or impairment of postnatal development [3]. Moreover, occupational exposure can also adversely affect reproductive health. For example, studies have shown that parental employment in agriculture may contribute to stillbirth and increase the risk of congenital malformations and organ-system defects in their offspring [4,5]. The nursing profession, or work as a hairdresser or cosmetologist have been associated with several adverse pregnancy outcomes [6,7,8,9,10].

Consequently, preventing exposure to environmental hazards is a priority as early as the preconception period, as recommended by the International Federation of Gynecology and Obstetrics in 2015 [11].

Several appropriate actions have already been implemented in both occupational and non-occupational settings. Preconception health care information to alert women about environmental risks during pregnancy is provided in many countries [12]. The dissemination of such information can take the form of individual counselling, group health education, mass media campaigns, or websites [12,13]. In the workplace, statutory measures have been implemented to prevent pregnant or nursing women from performing dangerous or unhealthy tasks in 111 countries throughout the world [14]. 

Yet, despite the existence of such preventive measures, it is unclear whether women planning or having a pregnancy are truly informed about the environmental risks to reproductive health and whether they subsequently change their behavior concerning their work practices or household habits, with possible help from health practitioners.

Although many existing studies have focused on knowledge and behavioral changes concerning ingested substances, such as alcohol, tobacco smoke, recreational drugs, or pharmaceuticals (especially folic acid and vaccines), only a few have dealt with other harmful environmental substances [15]. Two studies conducted in France showed that pregnant women do not receive sufficient information concerning the potential sources of exposure to five reprotoxic agents and endocrine disrupters [16,17]. However, these studies focused on targeted risks factors and did not investigate actual behavioral changes during pregnancy. 

In the workplace, one study carried out in France showed that, although exposure to occupational risk is frequent among pregnant workers, prevention is still very limited [18]. This study did not investigate the women’s perceptions concerning occupational risks or personal behavioral changes.

There are no studies that have jointly investigated occupational and non-occupational environments. However, it would be informative to simultaneously observe occupational and non-occupational environments, as they both contribute to the global exposure of women to reproductive hazards, with the proportion varying from case to case.

The first objective of our study was to evaluate women’s knowledge about the environmental exposures that could interfere with the proper course of pregnancy. The second objective was to assess the behavioral changes they established in their private and professional lives.

## 2. Materials and Methods

Our study was a cross-sectional survey of postpartum women hospitalized in the Department of Obstetrics and Gynecology of the Bordeaux University Hospital. This department contains a maternity ward that specializes in handling high-risk pregnancies and a neighborhood maternity for normal pregnancies. 

Our study included women hospitalized in the postnatal care and delivery unit during two periods (from 16 October 2017 to 26 October 2017 and 27 November 2017 to 7 December 2017) who gave birth to a healthy child. Indeed, the main preoccupation was to avoid generating anxiety among patients. We chose to exclude women who were currently pregnant, women with adverse pregnancy outcomes (such as preeclampsia, preterm delivery, intrauterine growth restriction, congenital malformation …), and those with language barriers or reading difficulties that prevented them from fully understanding the survey.

We developed an anonymous self-administrated survey composed of 30 items consisting of closed questions (available upon request from the authors). The survey was structured in four parts. Part 1 requested general information about the participants’ socio-demographic status and data concerning the pregnancy. Part 2 examined their perception of occupational risks to pregnancy and their attitudes at the workplace. In part 3, we asked them to rank 14 preventive measures that could be implemented at home in order of priority from 1 to 5 and asked them about behavioral changes they made in their personal life. Part 4 explored their information sources about the environment and pregnancy. 

The survey was distributed to the eligible women by a single trained midwife with a letter explaining the framework and purpose of the study. Help was offered to women who had difficulties understanding or answering the questions. Women who participated could return the survey to the perinatal health providers during their hospitalization.

The data were entered using Epi Info 7 software [19]. The same software was used for the analysis. Numeric variables are presented as means and medians and categorical variables as headcounts and proportions.

We then performed a comparison of the responses using the chi-squared test or Fisher’s exact test for various groups according to the following parameters: The education level defined by the Standard Classification of Education (ISCED) 2011 level (tertiary education corresponding to level 5 or more) [20], the declared number of pregnancies, and the possibility of exposure to environmental risk factors at work. Environmental risk factors on reproduction have been identified according to a method presented in a previous work [21]. We grouped them into four classes: Physical demands, reprotoxic agents, biological agents, and radiation. Following this, three experts categorized women in none or several categories of exposure on the basis of their profession and the company’s activity. Significance was defined as *p* < 0.05. This study was declared to the French CNIL (Commission nationale de l’informatique et des libertés, number N°2109748).

## 3. Results

### 3.1. Characteristics of the Study Population

In total, 121 hospitalized women responded to the survey (Figure 1).

Table 1 presents the main characteristics of our study population according to their work status during the pregnancy. Most women (*n* = 90, 74%) declared to have worked during this period. Workers tended to be older and had a higher education level than unemployed women (*p* = 0.02). In the total population, most women lived with a partner (92%), had planned the pregnancy (75%), and were aware of their pregnancy as soon as the first month (92%). 

The professions of women who worked during pregnancy are presented in order of frequency in Table 2, sorted according to the major groups defined by the International Standard Classification of Occupations (ISCO-08) [22].

### 3.2. Perception of Environmental Risks and Behavioral Changes at the Workplace

The nature of the job and the main activity of the company allowed us to categorize one woman (1.1%) as potentially exposed to ionizing radiation and nine (10.0%) to biological risk factors for reproduction, 20 (22.2%) as possibly exposed to reprotoxic agents, and 33 (36.7%) as potentially exposed to an accumulation of physical demands and/or organizational constraints.

There was a significant association between the level of education and exposure to occupational reproductive risk factors: Women with a high degree of education were less exposed at work than women with a lower level of education (*p* < 0.001).

Table 3 presents the behaviors and actions implemented by subjects at the workplace. Globally, 46% of the respondents (*n* = 39) declared their pregnancy to the employer during the first trimester of pregnancy and 55% stopped working during the last trimester. Only 10% (*n* = 8) consulted an occupational physician during the pregnancy. Nevertheless, their workstation was adapted during this period, 71% by themselves (*n* = 60) and 40% with the assistance of their employer (*n* = 34). Women considered to be exposed tended to be more inclined to establish preventive measures and stopped working earlier than unexposed workers, but the difference was not statistically significant. However, women with a lower level of education more often changed their working practices (*p* = 0.04). 

Figure 2 shows the behavioral changes initiated by the women themselves or their employers in the occupational environment. The most common measures implemented were adaptation of working hours and a reduction in biomechanical demands. 

There were no significant differences in behavior or the establishment of preventive measures at the workplace according to the women’s level of education or number of pregnancies.

### 3.3. Perceptions of Environmental Risks in the Non-Occupational Environment

Almost all women (91%, *n* = 107) considered the environment as a predominant factor to guarantee a healthy pregnancy. The answer was widely influenced by the level of education, as 99% (*n* = 68) of the more highly educated subjects responded favorably versus 80% among the women with a lower level of education (*p* = 0.001).

Figure 3 shows the five actions chosen by the subjects among 14 propositions to establish a healthy environment at home during pregnancy. We represented proportions of answers by order of priority (first choice or further selection). 

The most frequent preventive actions regarding lifestyle identified by 70% of women (*n* = 85) concerned limiting exposure to tobacco and alcohol. Answers to more specific questions concerning the potential link between alcohol and tobacco consumption and pregnancy issues are presented in Table 4 according to the subjects’ level of education. No respondents considered alcohol and tobacco consumption to be safe during pregnancy. Many women considered such consumption not only harmful during pregnancy but also during the preconception period. 

Thus, 56% (*n* = 29) of women with a lower level of education and 80% (*n* = 55) with a higher level of education indicated reducing or stopping tobacco and alcohol consumption as the most important actions to guarantee a safe pregnancy.

Changes in the consumption of alcohol were more frequently mentioned than for tobacco, especially among women with a higher level of education. They considered stopping alcohol consumption as essential to guarantying a healthy pregnancy (*p* = 0.05), some as even the most important action (*p* = 0.016).

### 3.4. Behavioral Changes in the Non-Occupational Environment

Behavioral changes in lifestyle concerning tobacco and alcohol consumption are presented in Table 5. In our study population, 18% of women (*n* = 21) smoked before pregnancy, with a higher proportion among women with a lower education level (*p* = 0.003). All smokers declared to have stopped or reduced their tobacco consumption at the beginning of the pregnancy. More women with a high level of education stopped smoking (*n* = 4, 64%), with no statistical difference. Ten women declared to have consumed alcohol daily before pregnancy and no subject declared to have consumed alcohol daily during pregnancy.

Changes in consumption habits among women are presented Figure 4. Subjects who did not ordinarily use the mentioned products were removed from the results. 

In our study population, 81% of women (*n* = 98) implemented at least one action among the 11 items proposed in the survey to reduce exposure to chemicals present in daily products. The median number of actions undertaken by the subjects was 3.5. More than the half the users reduced their consumption of hair dyes (73%), insecticide sprays (72%), mosquito repellent (60%), and home renovation products (59%). There were several differences concerning the actions implemented by women that approached the significance threshold, according to their level of education. Women with a higher level of education were more inclined to increase their consumption of organic food (*p* = 0.06) and decrease their use of hair dyes (*p* value = 0.05) and nail polish (*p* = 0.08). There was no statistical difference between these results depending on the number of pregnancies.

### 3.5. Source of Information about the Environment

Table 6 presents the frequency of women who considered they were sufficiently informed about the environment and subjects’ sources of information. Results have been ranked by priority order.

Internet and television were the first sources reported by 83% and 57% of women, respectively, whereas health practitioners were less frequently mentioned. Women indicated having more often been informed about the environment by midwives during their pregnancy than by other health professionals. Only 45% of the subjects (*n* = 54) were informed about environmental risks by a physician.

However, 82% of women (*n* = 68) considered that they were not sufficiently informed about environmental risks. They identified midwives, gynecologists, and obstetricians among the best professionals to advise them about environmental risks to pregnancy. Only 9% of women (*n* = 11) identified the occupational physician as a relevant source of information. 

## 4. Discussion

The main objective of our study was to evaluate the perception of women concerning the occupational and non-occupational environment and assess their behavioral changes during pregnancy.

Most (91%) women identified the environment as a major determinant of health during pregnancy. Although the risks concerning tobacco and alcohol consumption were well-known, the level of knowledge concerning other harmful substances was incomplete. Nevertheless, most women implemented preventive measures in both their occupational and non-occupational environment during pregnancy. At the workplace, most of such changes were made without the assistance of the employer or the occupational health physician. At home, most women implemented at least one action to diminish their exposure to chemical products. 

### 4.1. Perceptions of Environmental Risks and Behavioral Changes at Work

In our study, a large proportion of women (71%) implemented preventive measures by themselves, suggesting they were aware of the role of the environment during pregnancy. Fewer women (40%) benefited from preventive measures implemented by the employer. Nevertheless, we observed no statistical difference in the accommodations made between women considered to be exposed and those considered to be unexposed. This raises concerns about the relevance and adequacy of the adaptations implemented. 

According to the French Labor Code [23], the occupational physician is the key actor to assess risks at the workstation and propose appropriate accommodations, if needed. Based on our results, changes at the workstation were made without any contact with an occupational physician, as only 10% of the respondent women consulted with this practitioner during their pregnancy. These results can be explained by the absence of automatic notification by the employer to the occupational physician about the worker’s pregnancy, combined with the lack of early declaration. Indeed, 91% of women were aware of their pregnancy during the first month. However, they frequently delayed their declaration of the pregnancy to the employer until after the first trimester, even though this period corresponds to organogenesis, which is a window of high vulnerability during fetal development [24]. The workers’ lack of knowledge about the role of this practitioner could also explain the low frequency of consultations since only 9% of the subjects identified the occupational physician as a relevant source of information. Finally, several other barriers to effective prevention of pregnant and breastfeeding workers have been reported, such as the difficulty to identify substances that are dangerous for reproduction, the lack of updated criteria, guidance, and legislation on chemicals, and the lack of coordination among gynecologists and occupational physicians [25].

### 4.2. Knowledge and Behavior Concerning Alcohol and Tobacco

Our study showed that tobacco and alcohol are the best-known risk factors for pregnancy. These results are consistent with those of other studies that have been conducted in several other countries [15,26].

No women declared to have consumed alcohol daily during their pregnancy. The design of our survey did not permit us to record infrequent alcohol consumption. Among the respondents, 11% of women smoked during their pregnancy, but at lower levels than before. Despite the use of an anonymous survey, we cannot exclude that women underestimated their consumption of toxic substances (prevarication bias).

The proportion of tobacco and alcohol consumers in our population was lower than that revealed by another French study [16]. This can be explained by the higher proportion of women with a higher level of education in our population, as this parameter is associated with economic status, which has been identified as an important factor for substance use during pregnancy in the scientific literature [27]. 

Our study did not assess the possible reasons to explain the observed knowledge–behavior gap. Indeed, we observed that a good knowledge of potential risks was not sufficient to motivate women to stop smoking, in accordance with other studies [28]. Several quantitative studies have identified a number of barriers to smoking cessation among pregnant women, including willpower, the role and meaning of smoking, issues with the provision of smoking cessation aids, changes in relationship interactions, understanding of the facts, changes in smell and taste, and the influence of family and friends [28].

### 4.3. Knowledge and Behavior Concerning the Daily Use of Products

Almost all of the respondents declared to have implemented behavioral changes in their household habits. Most of the difference in knowledge and behavior among women were explained by the level of education and age. These results are consistent with those of another French qualitative study showing that these two parameters influence the perception of the risk due to endocrine disruptors in a population of pregnant women [17].

Household income may also explain some changes made in daily life, such as the increase in the consumption of organic foods and eco-labelled products. Indeed, in a French survey conducted by the Ministry in charge of the Environment, the respondents reported price as an important element in determining their consumption of labeled products, even if their perception of these products had improved over the past years [29].

### 4.4. Source of Information about the Environment

In our study, 70% of women considered that they were not sufficiently informed about environmental risks. Their major sources of information were the popular media (internet, television, and magazines), for which the quality of the information provided is highly variable. Few pregnant women were counselled about the environment by health providers. Several studies have shown that perinatal health providers do not generally ask patients about their environmental exposure, except for the consumption of toxic substances [30,31]. An American survey found that less than 20% of obstetricians routinely interview their patients about their environment [32]. The difficulties most frequently cited were the lack of training and knowledge in environmental health, the lack of evidence-based information, the short duration of consultations [30,31,32,33], the fear of patient reactions, and the lack of solutions [33].

Our results suggest that counselling by health professionals could be the most effective way to inform and encourage women to adopt appropriate behaviors during pregnancy. Respondents identified midwives, gynecologists, and obstetricians as the best sources of information. There is a true need to remove the barriers noted by physicians.

### 4.5. Strengths and Weaknesses of the Study

A strength of our study was the large scope of risk factors examined, notably harmful chemical substances that are rarely investigated in the scientific literature [15]. Moreover, we were able to simultaneously assess knowledge and behavioral changes, to identify potential gaps, and occupational and non-occupational environments, which are too often separated in medical approaches, preventing the establishment of global and effective preventive measures. Participation in the study was high (80%), suggesting a strong interest of the participants in this topic.

The characteristics of our study population were close to those of the French population of postpartum women in 2016 [34]. Overall, our results are largely consistent with the findings of other French studies, as already discussed [16,17,18].

A major limitation of our study was the exclusion of women who had adverse pregnancy outcomes or women with language barriers or reading difficulties. It is possible that this population may have been more exposed to environmental risk factors during pregnancy. The level of knowledge and the proportion of behavioral changes may be lower than among the women included in our study. Consequently, we cannot extrapolate our results to this specific population and dedicated studies should be conducted.

However, these results provide new information and contribute towards a better understanding of the perception and behavior of women concerning environmental risks for further reflection about the preventive measures that could be improved or implemented in occupational and household environments.

## 5. Conclusions

Women in this study perceive the environment as a strong factor that influences the proper course of pregnancy. Accordingly, a majority initiated behavioral changes during their pregnancy at work and/or in their private life. However, such corrective measures are rarely implemented in consultation with health practitioners, such as the occupational physician at the workplace or a perinatal practitioner in the non-occupational environment. Most of the information received by women comes from non-scientifically validated sources, such as the internet, television, and magazines. There is a true need to reduce the barriers identified in several studies to include the environment as a systematic component of pregnancy management, as suggested by the International Federation of Gynecology and Obstetrics [11]. 

## Figures and Tables

**Figure 1 ijerph-16-00565-f001:**
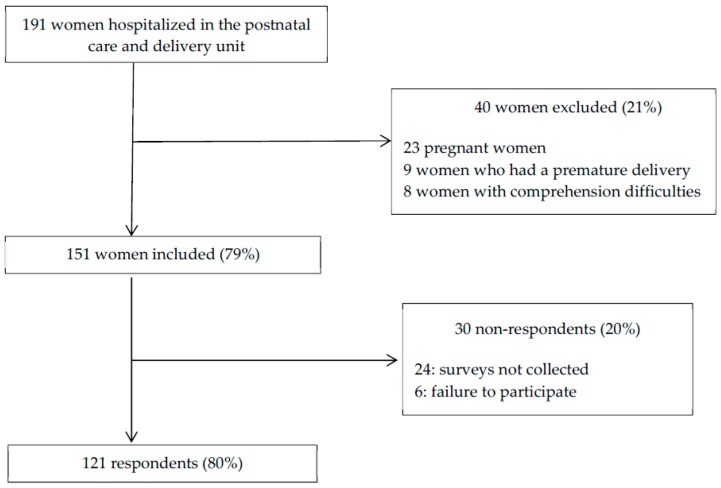
Inclusion and participation rate of women hospitalized in the postnatal care and delivery unit at the time of the study.

**Figure 2 ijerph-16-00565-f002:**
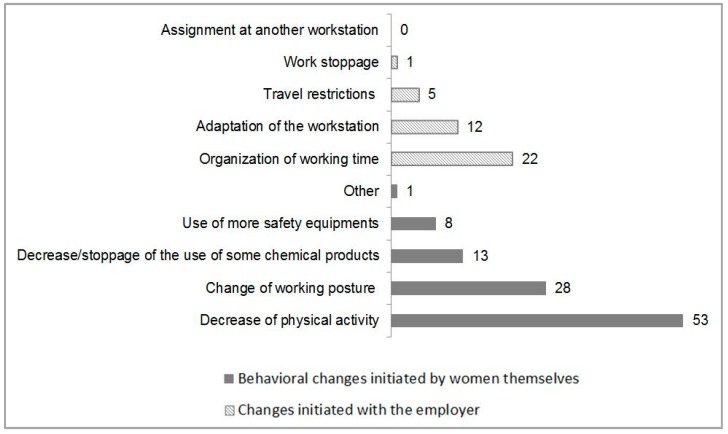
Behavioral changes initiated by women themselves or by their employers (*n* = 90).

**Figure 3 ijerph-16-00565-f003:**
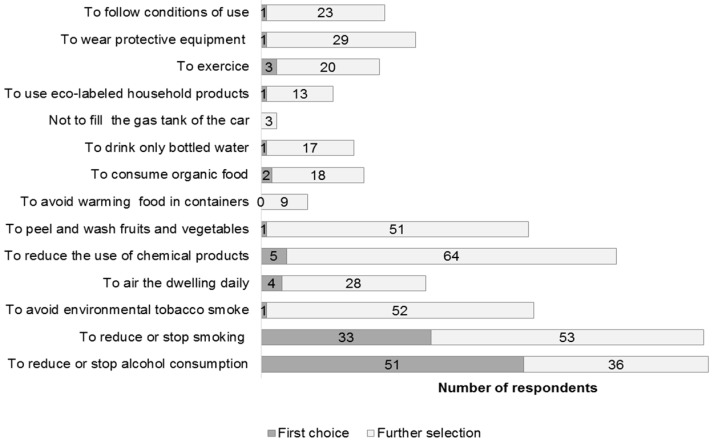
High priority actions chosen by women to improve their health during pregnancy (*n* = 121).

**Figure 4 ijerph-16-00565-f004:**
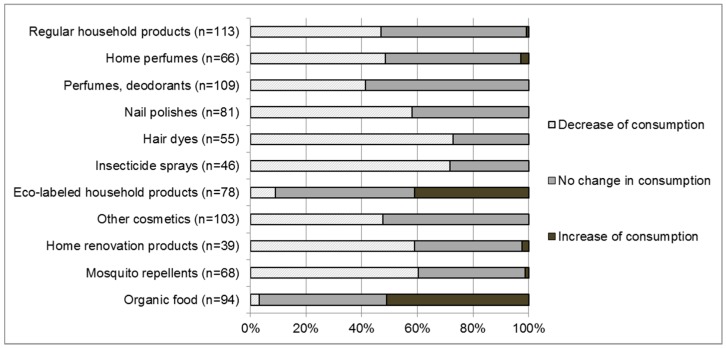
Change in consumption habits among women who use a range of common products (*n* = 121).

**Table 1 ijerph-16-00565-t001:** Description of survey respondents (*n* = 121).

	Total (*n* = 121)		Workers (*n* = 90)		Unemployed (*n* = 31)		*p*-Value *
	*n*	%	*n*	%	*n*	%	
Age							**0.03**
< 25	15	12.4	7	7.8	8	25.8	
25–35	69	57.0	52	57.8	17	54.9	
> 35	37	30.6	31	34.4	6	19.3	
Education level							**0.02**
Level 0 to 4	52	43.0	33	36.7	19	61.3	
Level 5 to 8	69	57.0	57	63.3	12	38.7	
Living with a partner ^1^	111	94.1	83	95.4	28	90.3	1
Planned pregnancy ^2^	91	75.2	70	77.8	21	67.7	0.28
Primigravid women	37	30.6	26	28.9	11	35.5	0.49
Knowledge of pregnancy ^3^							0.34
0–1 month	110	91.7	82	92.1	28	90.3	
1–3 months	9	7.5	7	7.9	2	6.5	
3–4 months	1	0.8	0	0	1	3.2	

^1^ Three missing values (two among the workers); ^2^ five missing values (two among the workers); ^3^ one missing value among the workers. * Chi-squared test or Fisher’s exact test.

**Table 2 ijerph-16-00565-t002:** Professions of women who worked during their pregnancy (*n* = 90).

Group of Profession	*n* (*n* = 90)	%
Technicians and associate professionals	30	33.3
Professionals	21	23.3
Service and sales workers	11	12.2
Managers	10	11.1
Clerical support workers	7	7.8
Elementary occupations	6	6.7
Skilled agricultural, forestry, and fishery workers	2	2.2
Total ^1^	87	96.6

^1^ Three missing values.

**Table 3 ijerph-16-00565-t003:** Actions implemented in the workplace during the pregnancy (*n* = 90).

	Total (*n* = 90)		Exposed (*n* = 42)		Unexposed (*n* = 48)		*p*-Value *
	*n*	%	*n*	%	*n*	%	
Pregnancy declaration date to the employer ^1^							0.75
0–3 months	39	46.4	19	51.4	20	42.6	
3–4 months	30	35.7	13	35.1	17	36.2	
4–6 months	13	15.5	4	10.8	9	19.1	
6–9 months	2	2.4	1	2.7	1	2.1	
Cessation of work ^2^							0.27
0–3 months	5	6.0	1	2.6	4	8.7	
3–4 months	6	7.1	4	10.5	2	4.3	
4–6 months	27	32.1	15	39.5	12	26.1	
6–9 months	46	54.8	18	47.4	28	60.9	
Behavioral change at the work station by women ^2^	60	71.4	30	78.9	30	65.2	0.17
Change made by the employer at the work station ^3^	34	40.5	13	34.2	21	45.7	0.29
No change made at the workstation	17	18.9	5	11.9	12	25	0.11
Medical visit with the occupational health physician ^4^	8	10.3	3	9.4	5	10.9	1

^1^ Six missing values or women without employer (five exposed and one unexposed subjects); ^2^ six missing values or women without employer (four exposed and two unexposed subjects); ^3^ six missing values (four exposed subjects and two unexposed subjects); ^4^ 12 missing values (ten exposed subjects and two unexposed subjects). * Chi-squared test or Fisher’s exact test.

**Table 4 ijerph-16-00565-t004:** Women’s knowledge about alcohol and tobacco consumption during pregnancy.

	Total (*n* = 121)		Level 0 to 4 of Qualification (*n* = 52)		Level 5 to 8 of Qualification (*n* = 69)		*p*-Value *
	*n*	%	*n*	%	*n*	%	
**Period of alcohol consumption dangerous for pregnancy ^1^**							0.78
As soon as the period of conception	28	23.7	11	22.4	17	24.6	
As soon as the beginning of pregnancy	90	76.3	38	77.6	52	75.4	
**Period of tobacco consumption dangerous for pregnancy ^2^**							0.23
As soon as the period of conception	54	46.2	19	39.6	35	50.7	
As soon as the beginning of pregnancy	63	53.8	29	60.4	34	49.3	
**Women who consider there to be no level of alcohol consumption that is safe for pregnancy ^1^**	116	98.3	48	98.0	68	98.6	1
**Women who consider there to be no level of tobacco consumption that is safe for pregnancy ^3^**	104	90.4	42	91.3	62	89.9	1
**First action considered by women to guaranty a healthy pregnancy**							
To reduce or stop alcohol consumption	51	42.1	13	25.0	38	55.1	**<0.001**
To cut down on or stop smoking	33	27.3	16	30.8	17	24.6	0.45
**Actions considered by women to guaranty a healthy pregnancy with no notion of ranking**							
To reduce or stop alcohol consumption	87	71.9	30	57.7	57	82.6	**0.002**
To cut down on or stop smoking	86	71.1	32	61.5	54	78.3	**0.04**

^1^ Three missing values (three subjects with a level 0 to 4 of qualification); ^2^ four missing values (four subjects with a level 0 to 4 of qualification); ^3^ six missing values (six subjects with a level 0 to 4 of qualification). * Chi-squared test or Fisher’s exact test.

**Table 5 ijerph-16-00565-t005:** Behavioral changes concerning alcohol and tobacco consumption during pregnancy (*n* = 121).

	Total (*n* = 121)		Level 0 to 4 of Qualification (*n* = 52)		Level 4 to 8 Qualification (*n* = 69)		*p*-Value *
	*n*	%	*n*	%	*n*	%	
Smoking status before knowledge of pregnancy ^1^							**0.003**
Non-smoker	98	82.4	35	70.0	63	91.3	
Smoker	21	17.6	15	30.0	6	8.7	
Daily consumption of cigarettes among smokers before knowledge of pregnancy							0.44
[1,2,3,4,5,6,7,8,9,10]	5	23.8	3	20.0	2	33.3	
[10,11,12,13,14,15,16,17,18,19,20]	11	52.4	7	46.7	4	66.7	
≥ 20	5	23.8	5	33.3	0	0.0	
Behavioral changes during pregnancy among smokers							0.15
Stopped smoking during pregnancy	8	38.1	4	26.7	4	66.7	
Cut down on smoking during pregnancy	13	61.9	11	73.3	2	33.3	
Daily consumption of cigarettes among smokers during pregnancy							0.58
[1,2,3,4,5]	4	30.8	4	36.4	0	0	
[5,6,7,8,9,10]	8	61.5	6	54.5	2	100	
[10,11,12,13,14,15]	1	7.7	1	9.1	0	0	
Daily number of alcohol units consumed before pregnancy ^2^							0.75
0	106	90.6	47	92.2	59	89.4	
1	11	9.4	4	7.8	7	10.6	
Daily number of alcohol units during pregnancy ^3^							1
0	118	100	51	100	67	100	

^1^ Two missing values (two subjects with a level 0 to 4 of qualification); ^2^ four missing values (one subject with a level 0 to 4 of qualification and three with a level 5 to 8 of qualification); ^3^ three missing values. * Chi-squared test or Fisher’s exact test.

**Table 6 ijerph-16-00565-t006:** Women’s sources of information about the environment (*n* = 121).

	Total	
	*n* (*n* = 121)	%
**Women who considered that they were sufficiently informed about the environment ^1^**		
No/Do not know	82	70.1
Yes	35	29.9
**Women who have been informed about the environment during pregnancy by physicians**	54	44.6
**Principal sources of information**		
Internet	100	82.6
Television	69	57.0
Midwife	52	43.0
Magazines	48	39.7
Entourage	44	36.4
Gynecologist	44	36.4
Maternity	30	24.8
General practitioner	30	24.8
**Relevant sources of information identified by women**		
Midwives	99	81.8
Gynecologists and obstetricians	88	72.7
General practitioner	45	37.2
Occupational physician	11	9.1
Other: Primary Sickness Insurance Fund	2	1.7

^1^ Four missing values.

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
