# Peer review of "Perception of Environmental Risks and Behavioral Changes during Pregnancy: A Cross-Sectional Study of French Postpartum Women"

_ijerph, 2019, doi:10.3390/ijerph16040565_

Round 1
Reviewer 1 Report
See attachment.

Author Response
We thank the reviewer for these valuable comments and suggestions. We made changes in the article accordingly to make the text clearer. The modifications are described hereafter, and we give a point-by-point response to the concerns.
Introduction
1. Page 1, Paragraph 1- Replace bromated with brominated
We changed bromated by brominated in the text.
2. Cigarettes are not generally ingested but rather inhaled.
We agree and we replaced “cigarettes” by “tobacco smoke” in the text.
3. It could argued that use of pharmaceutical or in some cases, prescription drugs have also been the subject of considerable study in relation to reproductive health and birth outcomes. Behavior associated with their use should at least be mentioned.
The review of Toivonen et al. 2017 also found that a lot of studies have targeted acid folic and vaccines in their assessment of women’s knowledge and/or behavior.
Consequently, we reworded the following sentence adding some information about pharmaceuticals:
“Although many existing studies have focused on knowledge and behavioral changes concerning ingested substances, such as alcohol, cigarettes, tobacco smoke, and recreational drugs or pharmaceuticals (especially folic acid and vaccines), only a few have dealt with other harmful environmental substances [15].”
We did not find other article dealing with this subject in the scientific literature.
Results
4. I suggest including a table with more detail regarding the exposure categories and possible exposure agents within categories.
We agree that more information is needed about the method. Consequently, we rephrased the following paragraph in the method :
“Environmental risk factors on reproduction have been identified according to a method presented in a previous work [21]. We grouped them into four classes. Four occupational exposure subgroups were determined by experts: physical demands, reprotoxic agents, biological agents, and radiation. Women were categorized by three experts on the basis of their profession and the company’s activity. Then, three experts categorized women in none or several categories of exposure on the basis of their profession and the company’s activity.”
5. Page 6, 8-Correct Error! Reference source not found. throughout. These appear to be references to figures.
The correction has been made as requested. “Error! Reference…” have been replaced by the figures ‘names in the text.
6. Figure 3-This is a confusing figure in terms of ranked priority. The text says these are ranked by priority; please clarify how that statement relates to the figure itself. The legend also states these are High priority actions. Were there low priority actions taken and how was priority ranking determined?
Information regarding the exercise of prioritization has been given in the part “Method”. “In part 3, we asked them to rank 14 preventive measures that could be implemented at home in order of priority from 1 to 5”.
Nevertheless, we agree that it is unclear and that we should also add information in the part “Result” to improve clarity of the text. We propose to rephrase the following sentence describing Figure 3’s content:
“Figure 3 shows the five actions chosen by the subjects among 14 propositions to establish a healthy environment at home during pregnancy, ranked by order of priority. We represented proportions of answers by order of priority (first choice or further selection).”
7. Page 9-Table 6 does not present levels of knowledge however that might be classified. It simply presents participants opinions or self-assessments of whether or not they felt sufficiently informed. Those are different from one another. It also presents self-reported sources of information. Please restate accordingly below the section 3.5 header to reflect that. For principal sources of information, were these reported as the first or most important source followed in some order ranking by other relevant sources or is that not correct? This could be explained more clearly and in a more detail in the methods section.
We agree with the reviewer’s comments. We have changed the presentation of Table 6 consequently:
“Table 6 presents the frequency of women who considered they are sufficiently informed about the environmental ant subjects’ sources of information of knowledge and the various sources of information concerning the environment mentioned by women. Results have been ranked by priority order.”
Discussion
8. Section 4.1-Are there any other reasons that occupational physicians may not be consulted? Are they always present and approachable? Are there any other reservations regarding occupational/job security that may influence participants willingness to consult occupational physicians? Please expand on these questions in the discussion.
In our study we identified three reasons explaining a lack of contact with an occupational physician that we described in the discussion (part 4.1. Perceptions of environmental risks and behavioral changes at work):
· the absence of automatic notification to the occupational physician by the employer about the worker’s pregnancy (In order to clarify we added the term “by the employer” in the corresponding sentence);
· a lack of early declaration (women can have stopped working before the occupational physician knows about the pregnancy;
· the workers’ lack of knowledge about the role of this practitioner.
Unfortunately, our study did not permit us to study other reasons that may influence participant’s willingness to consult occupational physicians. We did not find other studies that could have dealt with this question in the scientific literature. It would be interesting to investigate this question in further works.
9. Is it possible that some women may perceive that certain smoking cessation aids, especially chemical aids, are actually not actually safer than the tobacco products themselves?
Unfortunately, our study did not permit us to study the perception of impact aids ‘smoking cessation and the impact on the cessation. This should be the subject of a further work.
10. Space permitting, it would helpful if the authors could expand on the lack of an evidence base for many chemicals found in household products or that are released inside the home from other sources (e.g. chloroform from sanitized, treated residential water). This is one of the largest sources of uncertainty in clearly informing and educating the lay public, especially pregnant women. Environmental epidemiological information is thin to say the least and the requisite toxicology is lacking. Many of these chemicals have been understudied or have been the subject of study in more dated research using less sensitive measures. A few have been the dominant focus of contemporary study and it could be argued some of these may well be somewhat overrepresented. This is an opportunity to advocate for broadening the research to be more inclusive and holistic as well as to consider mixtures.
We totally agree with the importance to develop the research about chemicals found in daily products. This is a first step to permit authorities to properly inform the public, especially pregnant women, about household risks linked with the use of chemicals. This topic goes beyond the scope of our study but should be definitely addressed in another work.
Conclusions
11. Should lead this section off with Women in this study.
Accordingly to the reviewer’s comment, we changed the first sentence of the conclusions.
“Women in this study perceive the environment as a strong factor that influences the proper course of pregnancy.”
Reviewer 2 Report
Page 2 Materials and Methods, second paragraph: I would define what you mean by “adverse pregnancy outcomes” (at least provide examples) as this is rather vague.
Page 5 Results, Table 1: For clarity I would superscript the p-value and explain what test(s) were being used to compare which groups.
Page 6: “Error! Reference source not found. shows the behavioral changes initiated by the women hemselves or their employers in the occupational environment “
Page 6: “Error! Reference source not found. shows the actions chosen by the subjects to establish a ealthy environment at home during pregnancy, ranked by order of priority.”
Page 8: “Changes in consumption habits among women are presented in Error! Reference source not ound..”
Page 12: Just a note, but the other population that was excluded is also a limitation of the study
Author Response
We thank the reviewer for these valuable comments and suggestions. We made changes in the article accordingly to make the text clearer. The modifications are described hereafter, and we give a point-by-point response to the concerns.
1. Page 2 Materials and Methods, second paragraph: I would define what you mean by adverse pregnancy outcomes” (at least provide examples) as this is rather vague.
Accordingly to the reviewer’s comments, we add example to clarify the term: “adverse pregnancy outcomes”. Thus, we reworded the following sentence adding some information:
“We chose to exclude women who were currently pregnant, women with adverse pregnancy outcomes (such as preeclampsia, preterm delivery, intrauterine growth restriction, congenital malformation …), and those with language barriers or reading difficulties that prevented them from fully understanding the survey.”
2. Page 5 Results, Table 1: For clarity I would superscript the p-value and explain what test(s) were being used to compare which groups.
As proposed by the reviewer, we superscripted the significant p-value in bold and indicated the test performed by an asterisk *. We made changes at the Table 4 and Table 5 as well.
The tests have been also detailed in the last paragraph of the part “2. Materials and Methods”: “We then performed a comparison of the responses using the Chi Square Test or Fisher’s exact test […].Significance was defined as p < 0.05.”
3. Page 6: “Error! Reference source not found. shows the behavioral changes initiated by the women hemselves or their employers in the occupational environment “
Page 6: “Error! Reference source not found. shows the actions chosen by the subjects to establish a ealthy environment at home during pregnancy, ranked by order of priority.”
Page 8: “Changes in consumption habits among women are presented in Error! Reference source not ound..”
The correction has been made as requested. “Error! Reference…” have been replaced by the figures ‘names in the text.
4. Page 12: Just a note, but the other population that was excluded is also a limitation of the study
We agree with the reviewer. The exclusion of women with language barriers or reading difficulties is another limitation. We completed the following sentence consequently:
“A major limitation of our study was the exclusion of women who had adverse pregnancy outcomes or women with language barriers or reading difficulties.”